# Adolescents’ Alcohol Use: Does the Type of Leisure Activity Matter? A Cross-National Study

**DOI:** 10.3390/ijerph182111477

**Published:** 2021-10-31

**Authors:** Aranzazu Albertos, Ina Koning, Edgar Benítez, Jokin De Irala

**Affiliations:** 1School of Education and Psychology, University of Navarra, 31009 Pamplona, Spain; aalbertos@unav.es; 2Institute for Culture and Society (ICS), 31009 Pamplona, Spain; jdeirala@unav.es; 3Navarra Institute for Health Research (IdiSNA), 31008 Pamplona, Spain; 4Youth Studies, Interdisciplinary Social Science, Utrecht University, P.O. Box 80140, 3508 TC Utrecht, The Netherlands; i.koning@uu.nl; 5Instituto de Ciencia de los Datos e Inteligencia Artificial (DATAI), 31009 Pamplona, Spain

**Keywords:** adolescents, alcohol consumption, binge drinking, leisure, leisure activities, self-control

## Abstract

The main objective of this study was to analyze the relationship between structured, unstructured, and family leisure activities on the frequency of adolescent alcohol intake across three different countries (Spain, Peru, and The Netherlands). The self-control of adolescents was also investigated as a moderator in the relationship between leisure activities and alcohol consumption. Methodology: This research involved 4608 adolescents aged between 12 and 17 from three countries (Spain, Peru, and The Netherlands). In Spain and Peru, data was collected through a self-report questionnaire which was part of the Your Life project. In The Netherlands, a self-questionnaire was used, collected by the University of Utrecht. A multiple logistic regression was performed for each country. Results: The results showed that participation in unstructured leisure activities increased the likelihood of drinking more frequently and more heavily in all three countries. Structured leisure activities, in general, did not have a significant predictive effect on alcohol consumption in any of the countries. Family leisure activities reduced the risk of engaging in yearly alcohol use and yearly binge drinking among adolescents, especially in The Netherlands and Spain. The protective effect of family leisure and unstructured leisure risk on yearly alcohol use applied especially to Dutch adolescents with a low level of self-control. Discussion: The article emphasizes the need for parents to engage in leisure activities with their child; participation in unstructured activities is not to be encouraged.

## 1. Introduction

Adolescence is a stage in the life cycle in which young people start experimenting with the use of substances, such as the initiation of alcohol consumption. Adolescence is characterized by a period in time where youngsters are not concerned about the consequences of their long-term behavior but are more concerned with the short-term situation [1]. Alcohol consumption is a common risk behavior among adolescents [2], with numerous and serious effects on the lives of young people, which can affect their physical or psychological health [3].

Alcohol consumption (AC) is common among adolescents around the world, often with high levels of consumption [2,4]. However, there are differences between countries. On average, 58% of European and Canadian adolescents start experimenting with drinking alcohol before the age of 15. For example, 29% of Spanish adolescents and 26.5% of Dutch adolescents had their first drink before age 13 [5], whereas 50% of the adolescents in South America indicated that they drank alcohol before the age of 14 [4]. Once adolescents initiate drinking, many of them are involved in so-called binge drinking, i.e., drinking 5 or more glasses within a limited period of time. Nearly three quarters (70.8%) of Dutch youth < 16 years [6] and more than half (56%) of Spanish youth < 15 years [7] who had drunk alcohol in the previous month had been involved in binge drinking. Alcohol use during adolescence is related to several risks, related to physical, emotional, and social problems [8]. Particularly early and binge drinking increases the chance of having mental health problems and addiction to substances later in life [9,10,11]. It is important to investigate protective factors that may contribute to lowering drinking levels among youth [12], such as leisure activities.

In most industrialized countries, young people’s leisure time comprises about half of their waking hours [13,14]. The literature differentiates leisure activities between structured and unstructured leisure activities. Although not much research is available on the relative relations between specific types of leisure activity and the use of alcohol, there has been research conducted that investigated the role of unstructured leisure activities [15] or structured (school-based) activities [16,17]; there is much more known about the role of sports activities, specifically, in relation to drinking. Structured leisure (SL) is characterized by having a certain structure, a regular schedule, clearly defined goals and rules, a focus on building skills, and supervision by adults [18,19,20]. Overall, SL is considered beneficial for health and to support the positive development of youth [19,20,21]. In line with several studies, involvement in SL is expected to decrease involvement in alcohol use because spending time in structured activities, supervised by adults, lowers the opportunity to drink alcohol [22]. Research showed that adolescents who spend less time with their parents drink more alcohol than adolescents who spend more time with them [23]. Activities with parents can strengthen the social bond and can be considered a protective factor against alcohol use among adolescents [24]. Thus, it is highly likely that adolescents involved in supervised leisure are less likely to drink alcohol [12]. Moreover, it also decreases the possibility of these adolescents connecting with alcohol- and drug-using peers; for example, through increased participation in sports and clubs [25,26]. Unsupervised settings increase the likelihood of drinking alcohol [27] by increasing its accessibility [28].

Unlike SL, unstructured leisure (UL) is comprised of activities without a certain structure, regular schedule, or clearly defined goals and rules, and without adult supervision. Examples of UL are watching TV, surfing the internet, hanging out with friends, spending time in pool halls and bars, and going to shopping malls for fun or spending time in public places on a regular basis. These activities are associated with adolescent risk behaviors [29]. Participation in UL can be considered positive in certain unsupervised activities, with a strong socializing character among peers [30,31]. On the other side, according to numerous studies, time spent “hanging out” and lacking participation in organized activities predicts delinquency [32], behavioral problems, depression symptoms, poorer school grades [33,34], substance use [35,36], and participation in gambling [37]. Sharing time with peers (locally or at friends’ homes) is a form of UL that greatly increases situational incentives for problem behavior [38] and is associated with increased alcohol consumption [19]. In a study conducted by Chen et al. [39], participation in unstructured leisure activities predicted more frequent occasional drinking. In sum, the involvement of adolescents in UL activities is associated with a greater risk of alcohol use.

The differential susceptibility hypothesis stipulates that some adolescents are more susceptible to both positive and negative environmental effects than others, depending on, for example, individual characteristics [40], such as adolescents’ self-control. This indicates that the different types of leisure activities may influence adolescents’ alcohol use differently, depending on their level of self-control. The self-control of adolescents is related to a diversity of (risk) behaviors including alcohol use and overall well-being [41]. Moreover, several studies have demonstrated that adolescents with a low level of self-control are vulnerable to a variety of respectively risky and protective environmental factors such as parenting [42], digital media [43,44] and peers [45]. Therefore, it is likely that adolescents with a lower level of self-control benefit most from the protective environment (supposedly structured or family leisure activities) and are more vulnerable to the riskier environment (e.g., supposedly unstructured leisure).

The contribution of the type of leisure activity on drinking behavior among adolescents is not a well-studied area of research. Insight into the role of supervised and unsupervised leisure on adolescents’ drinking patterns can provide tools for prevention. Moreover, to our knowledge, there have been no cross-national studies conducted to investigate the role of leisure activities on adolescents’ alcohol use. The objective of this study is to evaluate the predictive effect of structured, unstructured, and family leisure activities on adolescents’ frequency of alcohol drinking across three different countries (Spain, Peru, and The Netherlands). In addition, this study also investigates moderation due to adolescents’ self-control in relation to leisure activities and alcohol use.

## 2. Materials and Methods

Data collected in three countries (Spain, Peru, and The Netherlands) were used in this study. Data obtained in Spain and Peru were collected in light of the same project—Your Life—[46]. Dutch data were collected in an independent project investigating alcohol use among adolescents in a community trial [47]. Therefore, in order to match data across projects, instruments were standardized based on the criteria of linguistic, semantic, and functional similarity [48].

### 2.1. Procedure

The data from Spain and Peru used in this study were collected as part of the Your Life Project, which was carried out in five countries, including Spain and Peru. The aim of this project is to know what young people feel and think about relationships, love, and sexuality. An invitation with information about the project was sent by e-mail to public and private high schools. In Spain, invitations were sent to all high schools available in public databases. In Peru, local collaborators include schools with easier access. In this way, the sampling presented is non-probabilistic.

Schools that agreed to participate were further informed about the process for the study. Adolescents in the participating schools were verbally informed about the purpose of the study; thereafter, written information was provided detailing the objectives of the project. Adolescent consent to participate in the study was formally established at the beginning of the online questionnaire [49]. Participants had the option to withdraw from the study at any time. Since Spain and Peru have different requirements regarding parental notification and consent, participating schools were informed of the different alternatives they had available to meet the requirements. The framework for requesting parental permission for research with adolescents is explained in Ruiz-Canela et al. [49].

The adolescent participants filled out a self-report questionnaire in which they did not provide names or any other personally identifiable information. Only the code of the school, the type of school, and the country were recorded in the database. The surveys were designed to ensure this anonymous participation [50]. To increase the respondents’ sense of privacy and promote honesty in reporting, questionnaires were administered at school, and teachers were instructed to not move around the computer room while the adolescents filled out the questionnaire.

The survey data were kept in a safe place at the project site. Access to the website is restricted to researchers directly involved in the study. The database linking the codes of the participating schools and their confidential information (name and address) was kept separate from the actual content of the survey. Data and results that allow for the identification of any school will never be published.

Study participants did not receive any incentive for their participation, but schools subsequently received overall results from their school, allowing them to use their results in specific educational programs, for the prevention of the problems identified through the survey. The design of the study was approved by the Ethics Committee of the University of Navarra, and each new participating center was asked to comply with the specific ethical requirements of the project. All relevant ethics committees in the countries participating in the project had access to the questionnaire prior to its application.

In The Netherlands, in light of a community intervention trial, the LEF program, data was collected by Utrecht University as commissioned by the municipality of Edam-Volendam [47]. For the LEF-project, two secondary schools in Edam-Volendam participated in the survey, including adolescents aged 12–17 years. Anonymous data was collected among adolescents with the passive permission of their parents. The adolescents were also given the opportunity to refuse participation and they provided active consent at the start of the online questionnaire. Participants filled out the questionnaire under an anonymous identification code, which could not be matched by the researchers involved in the study with any personal data. The online questionnaires were conducted during school-time in the school’s computer rooms under the supervision of a teacher and research assistant. The students were notified that the study would be about ‘factors that can stimulate the development of young people in a good and healthy way’. For this study, approval of the Faculty Ethical Review Committee from the Utrecht University (FETC18-060) was obtained.

### 2.2. Participants

For Spain and Peru, the data were gathered during the period between December 2016 and September 2018. In total, 18 schools for Spain and 15 for Peru participated in the study, to which the link to the website was provided, designed to provide detailed information to the participants (https://proyectoyourlife.com//, accessed on 16 August 2021). The surveys were completed in the classroom under the supervision of the teacher in charge, in all its stages. At the time of this work, the sample was made up of 3228 persons. After evaluation for inconsistency, incomplete surveys, and requirements on the variables, the sample was reduced to 2506 respondents, distributed in 1276 (50.92%) from Peru and 1230 (49.08%) from Spain. For both countries, the age of the respondents was between 12 and 17 (mean age = 14.28, SD = 1.35) and 62.4% were female and 37.6% were male.

The Dutch data were collected in May 2018. In The Netherlands, schools that offer all levels of education often have more than 1000 students. In the participating schools, 2166 adolescents filled out the questionnaire. The data was controlled for inconsistency, incomplete surveys, and the requirements on the variables; therefore, 64 respondents were excluded from the dataset. A remaining 2102 respondents between the ages of 12 and 17 participated (mean age = 14.7, SD = 1.32). More than half (52.2%) were female and 47.8% were male.

### 2.3. Measures

The assessment instrument used in Spain and Peru was the Your Life project self-report questionnaire [46], with three different age-related versions (13, 15, and 17 years). As a reference source for the creation of our questionnaire we used the Illustrative Questionnaire for Interview-surveys with Young People [51]. The questionnaire was first written in Spanish and later adapted to English.

For the scales of yearly alcohol use (YAU) and yearly binge drinking (YBD), participants were asked to report the number of drinking occasions where, for YAU, they consumed at least one alcoholic beverage during the last year, and, for YBD, occasions where they consumed 5 or more alcoholic drinks within a few hours in the last year [52]. For Spain/Peru, responses were on a five-point scale (0 = Never, 1 = Less than 1 day per month, 2 = 1–3 days a month, 3 = 1–2 days a week, 4 = 3 days a week or more). For The Netherlands, response options were on a 14-point scale (1 = zero, 2 = 1 … 10 = 9, 11 = 10, 12 = 11 to 19, 13 = 20 to 39, 14 = 40 or more). Finally, both instruments were dichotomized in never and any YAU or YBD (Table 1).

Structured leisure (SL) indicates activities that adolescents engage in under the supervision of adults, those with a certain structure, regular schedule, clearly defined goals and rules, and focused on building skills [18]. The instrument developed by Carlos et al. [46] was used to ask about engagement in a variety of activities, both structured and unstructured. Adolescents were asked how often they engaged in three different types of activities: voluntary work, artistic or cultural activities, and exercising. A five-point Likert scale was used: 1 = Every day, 2 = Multiple days a week, 3 = Approximately one day a week, 4 = Less than one day a week, 5 = Never. The responses were averaged, and these values were assigned to ordered tertiles (three groups of approximately 33% of individuals), indicating a low, middle, and high frequency of participation in SL activities, split by countries.

Family activities indicate activities that adolescents reported they engaged in with their parent(s). Adolescents were asked how often they engaged in eight different activities (The Netherlands, [53]) or four different activities (Spain and Peru) together with their parent(s), e.g., eating together or going somewhere together. Each item could be answered on a five-point Likert scale: 1 = Never, 2 = Less than 1 day a month, 3 = 1–3 days a month, 4 = 1–2 days a week, 5 = 3 days a week or more. The responses were averaged, and these values were assigned to ordered tertiles (three groups of approximately 33% of individuals), indicating low, middle, and high frequency of participation in family activities, split by countries.

Unstructured leisure (UL) was measured by asking about adolescents’ involvement in two (The Netherlands) or three (Spain and Peru) different types of activities, e.g., meeting in a place with friends, without adults present (0 = Never, 1 = Less than 1 day a month, 2 = 1–3 days a month, 4 = 1–2 days a week, 5 = 3 days a week or more). The responses were averaged, and these values were assigned to ordered tertiles (three groups of approximately 33% of individuals), indicating low, middle, and high frequency of participation in UL activities, split by countries.

Self-control [54] (SC) was measured through the evaluation of three behaviors (see Table 1), using five-point Likert-type scales. For The Netherlands, the response options were 1 = Never, 2 = Rarely, 3 = Occasionally, 4 = Often, and 5 = Very often. For Spain and Peru, the response options were 1 = Never, 2 = Almost never, 3 = Sometimes, 4 = Almost always, 5 = Always. The responses were averaged, and these values were assigned to ordered tertiles (three groups of approximately 33% of individuals), indicating low, middle, and high average frequency of SC behaviors, split by countries.

The possible confounders were analyzed as follows: sex was dichotomized in females (0) and males (1); age group (AG) was categorized into early and late adolescents, following the median value of the population (14.3 years old); family status was assigned according to whether the child lived with one (0) or both parents (1). Socioeconomic status was self-reported in The Netherlands, and for the remaining countries it was reported by the school (schools indicated whether the majoritarian status of their families was high, average, or low income). However, the lower socioeconomic level was highly underrepresented in the sample, especially for Spain, which was 2.4%, or in The Netherlands, which reached only 5.4%. Therefore, it was decided to merge this level with the medium level and to only consider two values for this variable: low and high socioeconomic level.

### 2.4. Analysis

Initially, the data were evaluated through descriptive statistics, and then an association test for YAU and YBD was performed with all the variables by an χ^2^ test.

The proposed model considers reports of YAU and YBD predicted by the three types of leisure (structured, unstructured, and familiar) and self-control, corrected for sex, age group, family situation, and the socioeconomic level. Consequently, the proposed statistical model of analysis was a multiple logistic regression for each country. The evaluation of multicollinearity between the predictors was evaluated using the statistics of variance inflation factor, tolerance, and condition index. In all cases, complete or quasi-complete separation was evaluated together with diagnostic statistics for each individual observation: predicted values, residuals, and influence. Odds ratios were estimated with and without (raw) confounders and evaluating interactions between the three types of leisure. *Alpha* was settled to α = 0.05. The analyses were performed with SAS v. 9.4.

## 3. Results

### 3.1. Descriptive Statistics

The main characteristics of the participants are presented in Table 2. More than half of the participants were female (59%). In The Netherlands, 57.9%, of the participants were between 15 and 17 years old, while in the other two countries the majority were between 12 and 14 years old (58.5% in Peru and 55.2% in Spain). In all three countries, more than 80% of the young people involved had both parents and more than 60% had lower socio-economic status. Peruvian and Spanish adolescents showed a higher level of self-control (34.8% and 35.7%, respectively) compared to Dutch adolescents (28.6%).

### 3.2. Association Tests

Table 3 presents the results of the tests for the association between predictors and the use of Yearly Alcohol Use (YAU) and Yearly Binge Drinking (YBD) without correction by confounders. It is observed that, for both YAU and YBD, their prevalence was highest in The Netherlands (45.7% and 31.6%, respectively) and lowest in Peru (21.4% and 8.6%). As for the sex of the respondents, no associations with the use of both alcohol outcomes were observed (*p* = 0.071 and *p* = 0.057, in both data sets). In terms of the age group of respondents, YAU was about four times higher in the older age group compared to the younger age group. For YBD, the ratio rose by nearly 7 times. By relating two-parent families with one-parent families, a higher prevalence was observed, for both YAU and YBD, in single-parent families. Finally, for sociodemographic variables, the socioeconomic level was only associated with the use of YBD, where this practice was more prevalent for high socioeconomic levels.

Regarding the behavioral variables, the protective effect of self-control and family leisure was observed for both YAU and YBD. This is contrary to the UL, which consistently shows its effect as a risk factor for these types of alcohol consumption. A different result was obtained for SL, which only showed an effect on the use of YAU, with the intermediate levels for those with the highest prevalence of use.

The models evaluated for alcohol consumption, in general, showed the same behavior in the three countries (Table 4). The protective effect of having two parents was highlighted in all countries (with and without adjustment for confounders). The highest frequency of alcohol consumption was in late adolescence, with an OR between 5.4 and 9 in comparison to early adolescence. The effect of alcohol consumption as a function of the level of UL was also consistent, with the OR between 2.1 and 3.3 when comparing low levels of UL with medium levels, and between 3.2 and 6.2 when comparing low levels with high levels. As expected, family leisure was confirmed in the three countries as a protection factor, especially the high levels of family leisure with obtained ORs between 0.40 and 0.43 compared with low levels of this variable. In Peru, the average levels of family leisure were not enough to be protective when compared to the low levels, contrary to the other two countries. On the other hand, SL, in general, did not protect against alcohol consumption in the three countries; there was an exception in Spain, where intermediate levels of this type of leisure were a risk factor. Although no significant OR values were detected, the estimates for Peru and Spain showed values greater than one. Income was identified as a risk factor only in Peru with an OR of 2.14 in relation to lower income levels. Finally, the sex of the respondent did not show a significant relationship with alcohol consumption in Peru and Spain.

Regarding YBD (Table 5), there was a response similar to YAU, with a protective effect of family leisure or having both parents, and unstructured leisure and belonging to the late adolescent age group as risk factors. In Peru, high income was a risk factor, and the average levels of family leisure did not protect against this type of consumption. There was no relationship between structured leisure and YBD in all countries. Regarding differences, in relation to the YAU, it was found that for Spain having two parents did not have a significant protective effect. It is also noteworthy, for this type of consumption, that the ORs by sex already showed females with risk values in relation to males in The Netherlands (1.47). ORs associated with structured leisure were not significant (*p* > 0.05).

When evaluating the model including the interactions of self-control with the different types of leisure, the presence of interaction for The Netherlands was observed with unstructured and family leisure (Table 6). In the case of unstructured leisure (Figure 1), people with a low level of unstructured leisure benefited from medium and high levels of self-control (w = 10.27; *p* = 0.04). In other words, if the level of self-control is low, the probability of alcohol consumption increases despite not being exposed to medium or high levels of unstructured leisure. Regarding the interaction between family leisure and self-control (Figure 2), it is noted how children with low levels of self-control especially benefited from increases in the levels of family leisure—even at the highest levels of family leisure—since this condition manages to neutralize the negative effects of the lack of SC (w = 13.75; *p* = 0.01) (Table 6).

## 4. Discussion

This study investigated the predictive effect of structured (SL), unstructured (UL), and family leisure activities on the frequency of annual alcohol consumption (YAU) and excessive alcohol consumption (YBD) of adolescents in three countries (Spain, Peru, and The Netherlands). In addition, this study also investigated moderation due to adolescents’ self-control in relation to leisure activities and alcohol use. Findings demonstrated that, in line with our hypothesis, involvement in UL activities increased the likelihood of drinking alcohol more frequently and intensively (binge drinking) across all three countries. However, SL was not negatively associated with alcohol use in any of the countries. Moreover, family leisure activities only lowered the risk of involvement in yearly and binge drinking among adolescents in The Netherlands and Spain, not in Peru. The protective association between family leisure and the risk of UL on YAU (not binge drinking) particularly applied to Dutch adolescents with a low level of self-control.

For both alcohol outcomes (YAU and YBD), family leisure could be considered to be a protective effect, whereas unstructured leisure generally appeared to be a risk factor. These results are in line with those obtained in another study [35], in which no protective effect of extracurricular activities on the initiation of alcohol use was found, while playing sports was associated with a decrease in the likelihood of initiating marijuana and tobacco use. In contrast, they differ from those found by other studies, in which SL activities had a significant protective effect against alcohol use in young people [22,55,56]. The protective effect of engaging in leisure activities with parents has also been found by other studies [23]. According to Barnes et al. [24], family time is a protective factor against problem behaviors, whereas time spent with peers is a risk factor for problem behaviors. In addition, access to alcohol may also be related to family leisure activities as a protective effect and to unstructured leisure as a risk factor. Thus, in three countries within and outside Europe, the importance of young people engaging in family leisure activities was shown, while unstructured leisure activities could be considered a risk factor for adolescent drinking. [22].

Though a higher level of self-control is associated with a lower level of drinking across all three countries (except for YBD in Peru), only in The Netherlands did adolescents’ self-control influence the relation between family leisure and UL on YAU (not binge drinking). That is, the positive character of family leisure and the risk of unstructured leisure on yearly alcohol use was strongest among Dutch adolescents with a low level of self-control. This implies that, in The Netherlands, parents spending time with their kids should be encouraged, and involvement in unstructured activities should be discouraged—particularly among adolescents with a low level of self-control; this subgroup is particularly vulnerable to involvement in a variety of risk behaviors [57]. According to Hofstede’s rating [58], The Netherlands is considered an individualist society which is characterized by people who are supposed to look after themselves and their direct family only. Spain and Peru are more collectivist societies where people belong to ‘in-groups’ that take care of them in exchange for loyalty [59]. This may indicate that in more individualistic countries, such as The Netherlands, self-control plays a more important role in adolescents’ behavior than in countries where a more collectivistic, familial structure is present. It seems that in individualistic societies, it is even more relevant for parents to do activities together with their child, and to discourage the involvement of their kids in activities that are unsupervised and unstructured. Parents should be made aware of the importance of doing activities together with their child. In addition, involvement in unstructured activities should not be promoted—particularly in more individualistic countries where there are at-risk adolescents, i.e., those with a lower level of self-control, who are more vulnerable to drinking alcohol.

Some comments are worthy concerning the response rates in our analysis. The Your Life project is not a representative survey, being more akin to a convenience sample, where response rates are not relevant. In such studies, representativeness is not a goal and thus the response rate is not an issue, but follow up rates are. Since the analysis we are presenting does not entail follow up data, the presence of confounding factors is addressed by multivariate adjustment, so large sample sizes are more important, and this goal has been achieved in this study. The Netherlands data has an estimate of the response rate because the initial target sample consisted of two schools. This was thus a closed sampling frame. However, also in their case it was more important to adjust for confounding factors, as representativeness was not an objective either.

This study has some limitations. The cross-sectional design does not necessarily allow the direction of the association to be described. However, these results link unstructured activities as a risk factor for YAU and YBD drinking, and family leisure activities as a protective factor. The responses of the questionnaires are self-reported, but social desirability should be minimized using anonymous questionnaires.

Despite these limitations, this study has several strengths. First, the large sample size allowed for adjustment for possible important confounding factors and allowed interactions to be assessed. Second, the association of YAU and YBD with SL and UL was analyzed, along with each specific leisure activity and with self-control.

## 5. Conclusions

Adolescents’ participation in UL activities increases the likelihood of drinking alcohol more frequently in all three countries (YAU and YBD). Family leisure had a protective effect, whereas unstructured leisure was presented as a risk factor for drinking. A higher level of self-control predicted a lower level of alcohol consumption in all three countries. In The Netherlands, a difference was found, where the protective effect of family leisure and the risk of unstructured leisure on annual alcohol consumption was stronger among adolescents with low self-control.

## Figures and Tables

**Figure 1 ijerph-18-11477-f001:**
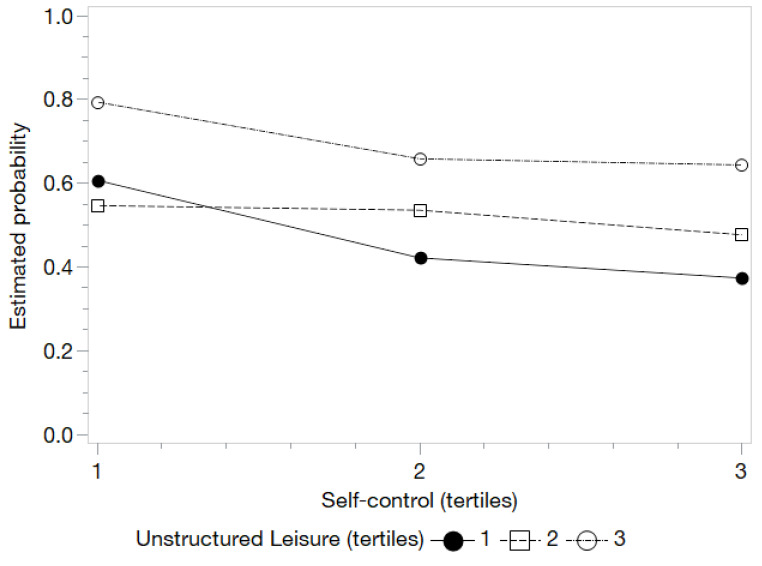
Estimated probabilities for the interaction between self-control and unstructured leisure for the YAU in The Netherlands.

**Figure 2 ijerph-18-11477-f002:**
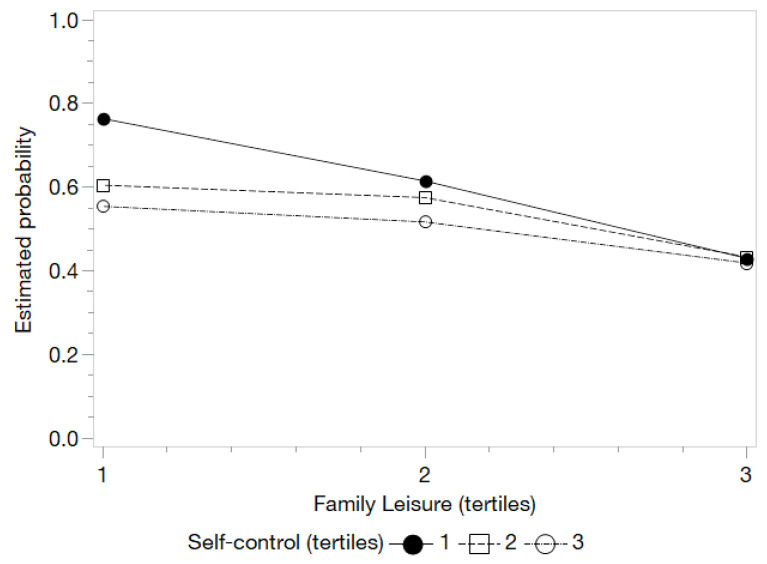
Estimated probabilities for the interaction between self-control and family leisure for the YAU in The Netherlands.

**Table 1 ijerph-18-11477-t001:** Pairing by linguistic, semantic, and functional similarity for instruments of Your Life and Utrecht projects (questions and answer options).

The Netherlands	Spain/Peru (Your Life ^1^)	Final Paring
Alcohol use past 12 months (times)	Alcohol use past 12 months	Yearly alcohol use (YAU)
0	Never	Never (0)
1	Less than one day a month	Any use (1)
2–9	1–3 days a month	
10 or more	3 or more days a week	
Binge drinking past 12 months (times)	Binge drinking past 12 months	Yearly binge drinking (YBD)
0	Never	Never (0)
1	Less than one day in a month	Any event (1)
2–9	1–3 days in a month	
10 or more	3 or more days in a week	
Family situation	Marital status of parents	Family situation (FS)
Yes, both parents	They have never married (each other)	Both parents (1)
	Married	
No, my parents are divorced	Separated/divorced, but neither has remarried, nor has a stable partner	Just one parent (0)
No, my father passed away		
No, my mother passed away		
Other, namely…	Other	Missing
Socioeconomic status of the family	Socio economic status of the school	Socioeconomic status (SES)
Very rich	High income	High income
Quite rich		
Average	Average income	Average income
Not that rich	Low income	Low income
Not rich at all		
Self-Control	Self-Control	Self-Control (SC)
I’m good at working towards goals far into the future.	I plan the things I do.	Tertiles of the average of the items.
Having fun leads to me not finishing my work.	I usually finish what I start.	
I often do things without thinking them through in advance.	I do things without thinking.	
Family leisure	Family leisure	Familiar leisure (FL)
Playing games inside together (e.g., board games, computer games).	We do sports, outings, or excursions, play board games.	Tertiles of the average of the items.
Going somewhere together (e.g., to the city, the beach, shopping, sports matches).	I usually dine with my parents.	
Exercising together.	They talk with you [your parents] about your interests (your hobbies, the things you like…)	
Going for a walk together.	[Your parents:]—They have time to talk to you.	
Eating together.		
Talking about things together.		
Visiting family or friends together.		
Watching television, a movie, or a series together.		
Structured leisure	Structured leisure	Structured leisure (SL)
Voluntary work (e.g., at a sports club or an institution).	Volunteering (collaborating with an NGO, charity, etc.)	Tertiles of the average of the items.
Artistic or cultural activities (e.g., museum/theatre visits, making music, acting, or painting).	Making things together or attending artistic and formative activities (music, painting, theater, courses, talks, catechesis, etc.)	
Exercising (e.g., football/soccer, tennis, hockey, or fitness).	Participating in sports, going to the mountains, etc.	
Unstructured leisure	Unstructured leisure	Unstructured leisure (UL)
Hanging out with friends at home, without supervision of parents.	Hang out on the street, in a park, on the beach, or in other public places.	Tertiles of the average of the items.
Other leisure activities (e.g., shopping, visiting sports matches, playing pool).	Meeting in a place where you are only the group of friends, without adults present.	
	Hanging out in shopping centers, games rooms, billiards, soccer stadium, etc.	

^1^ Original version in Spanish.

**Table 2 ijerph-18-11477-t002:** Characteristics of the sample.

Characteristic	The Netherlands N = 2102 %	Peru N = 1276 %	Spain N = 1230 %
Sex			
Male	47.8	39.7	35.6
Female	52.2	60.3	64.4
Age group			
Early adolescents	42.1	58.5	55.2
Late adolescents	57.9	41.5	44.8
Family structure			
One parent	17.8	13.1	14.2
Both parents	82.2	86.9	85.8
Socioeconomic level			
Low	66.4	63.1	73.8
High	33.6	36.9	26.2
Self-control			
Low	38.2	26.7	24.8
Middle	33.2	38.5	39.5
High	28.6	34.8	35.7
Structured leisure			
Low	20.2	30.7	25.3
Middle	47.8	32.1	36.2
High	32.0	37.2	38.5
Unstructured leisure			
Low	26.0	38.0	36.9
Middle	44.2	37.2	38.2
High	29.8	24.8	24.9
Family leisure			
Low	35.0	27.5	37.9
Middle	32.3	37.7	23.2
High	32.7	34.8	38.9
Yearly alcohol use			
Never	54.3	78.6	66.3
Any	45.7	21.4	33.7
Yearly binge drinking			
Never	68.4	91.4	85.7
Any	31.6	8.6	14.3

**Table 3 ijerph-18-11477-t003:** Demographics and behaviors, by alcohol and binge drinking use. Association test (χ^2^) between the levels of those factors and YAU and YBD.

Item	N	Yearly Alcohol Use (%) ^1^		Probability of χ^2^ test	Yearly Binge Drinking (%) ^2^		Probability of χ^2^ Test
		No	Any		No	Any	
Demographics							
Country							
The Netherlands	2102	54.3	45.7	<0.001	68.4	31.6	<0.001
Peru	1276	78.6	21.4		91.4	8.6	
Spain	1230	66.3	33.7		85.7	14.3	
Sex							
Male	1949	65.7	34.3	0.071	80.7	19.3	0.057
Female	2659	63.1	36.9		78.4	21.6	
Age group							
Early adolescents	2309	85.2	14.8	<0.001	94.7	5.3	<0.001
Late adolescents	2299	43.2	56.8		64.0	36.0	
Family structure							
One parent	717	54.1	45.9	<0.001	72.0	28.0	<0.001
Both parents	3891	66.1	33.9		80.8	19.2	
Socioeconomic status							
Low	3109	65.1	34.9	0.089	80.2	19.8	0.044
High	1499	62.5	37.5		77.7	22.3	
Behaviors ^3^							
Self-control							
Low	1448	56.1	43.9	<0.001	73.3	26.7	<0.001
Average	1675	64.4	35.6		79.7	20.3	
**High**	1485	72.1	27.9		85.0	15.0	
**Structured leisure**							
**Low**	1127	67.2	32.8	<0.001	79.8	20.2	0.140
**Average**	1860	60.7	39.3		78.0	22.0	
**High**	1621	66.3	33.7		80.7	19.3	
**Unstructured leisure**							
**Low**	1486	80.2	19.8	<0.001	91.3	8.7	<0.001
**Average**	1874	60.8	39.2		76.9	23.1	
**High**	1248	50.4	49.6		68.8	31.2	
**Family leisure**							
**Low**	1553	53.6	46.4	<0.001	28.3	27.2	<0.001
**Average**	1445	64.4	35.6		19.8	19.1	
**High**	1610	74.3	25.7		13.9	13.4	

^1^ Refers to any consumption of alcohol. ^2^ Corresponds to consumption of five or more alcoholic drinks within a 2-h period. ^3^ Categorical ordinal variables of the inventory, related to the factor, became tertiles for each country.

**Table 4 ijerph-18-11477-t004:** Odds ratio estimated and 95% confidence limits for yearly alcohol use (YAU) in each country, raw and adjusted by confounders.

Independent Variables	The Netherlands		Peru		Spain	
	OR (CI 95%)	Adj. OR (CI 95%)	OR (CI 95%)	Adj. OR (CI 95%)	OR (CI 95%)	Adj. OR (CI 95%)
**Female (Ref = male)**	**1.28 (1.08–1.52) ^1^**	1.18 (0.97–1.45)	1.18 (0.89–1.55)	1.34 (0.96–1.87)	1.11 (0.87–1.43)	1.26 (0.91–1.74)
**Both parents (Ref = one parent)**	**0.70 (0.56–0.87)**	**0.72 (0.56–0.93)**	**0.57 (0.40–0.82)**	**0.66 (0.43–1.00)**	**0.57 (0.41–0.79)**	**0.57 (0.39–0.85)**
**High income (Ref = low income)**	1.08 (0.90–1.29)	1.19 (0.96–1.47)	**2.22 (1.69–2.91)**	**2.14 (1.54–2.98)**	**0.68 (0.52–0.90)**	1.19 (0.83–1.72)
**Late adolescence (Ref = early adolescence)**	7.11 (5.81–8.70)	6.88 (5.58–8.49)	5.85 (4.33–7.91)	5.40 (3.91–7.47)	8.86 (6.72–11.68)	9.00 (6.65–12.19)
**Structured leisure (Ref = low level)**						
High	0.93 (0.73–1.19)	0.95 (0.72–1.27)	**1.47 (1.04–2.06)**	1.19 (0.79–1.77)	0.77 (0.57–1.05)	1.01 (0.70–1.46)
Middle	0.96 (0.76–1.20)	1.01 (0.78–1.32)	**1.48 (1.05–2.10)**	1.47 (0.98–2.20)	1.28 (0.95–1.74)	**1.51 (1.05–2.18)**
**Unstructured leisure (Ref = low level)**						
High	**2.72 (2.14–3.45)**	**3.2 (2.43–4.23)**	**7.84 (5.22–11.8)**	**6.21 (3.98–9.71)**	**4.15 (3.01–5.73)**	**4.57 (3.13–6.66)**
Middle	**2.12 (1.70–2.65)**	**2.10 (1.63–2.70)**	**3.98 (2.67–5.94)**	**3.28 (2.14–5.03)**	**2.29 (1.70–3.09)**	**2.18 (1.55–3.07)**
**Family leisure (Ref =low level)**						
High	**0.40 (0.32–0.50)**	**0.41 (0.32–0.53)**	**0.45 (0.31–0.64)**	**0.43 (0.28–0.66)**	**0.38 (0.28–0.50)**	**0.39 (0.28–0.55)**
Middle	**0.57 (0.46–0.70)**	**0.58 (0.45–0.73)**	0.94 (0.68–1.28)	0.87 (0.60–1.26)	**0.67 (0.50–0.91)**	**0.60 (0.42–0.86)**

^1^ Bold number show significant OR with *p* < 0.05.

**Table 5 ijerph-18-11477-t005:** Odds ratio and 95% confidence limits for yearly binge drinking (YBD) in each country, raw and adjusted by confounders.

Independent Variables	The Netherlands		Peru		Spain	
	OR (CI 95%)	Adj. OR (CI 95%)	OR (CI 95%)	Adj. OR (CI 95%)	OR (CI 95%)	Adj. OR (CI 95%)
**Female (Ref = male)**	**1.51 (1.26–1.82) ^1^**	**1.47 (1.19–1.83)**	1.12 (0.75–1.67)	1.05 (0.67–1.65)	0.96 (0.69–1.34)	0.85 (0.57–1.27)
**Both parents (Ref = one parent)**	**0.70 (0.56–0.88)**	**0.72 (0.55–0.94)**	**0.47 (0.29–0.76)**	**0.55 (0.32–0.93)**	0.67 (0.44–1.02)	0.79 (0.49–1.25)
**High income (Ref = low income)**	1.20 (0.99–1.46)	**1.38 (1.09–1.73)**	**2.12 (1.43–3.14)**	**1.88 (1.20–2.95)**	**0.64 (0.43–0.95)**	0.86 (0.53–1.40)
**Late adolescence (Ref = early adolescence)**	**10.67 (8.17–13.91)**	**10.45 (7.94–13.74)**	**9.84 (5.71–16.93)**	**8.53 (4.89–14.9)**	**6.33 (4.28–9.35)**	**5.67 (3.76–8.54)**
**Structured leisure (Ref = low level)**						
High	0.92 (0.71–1.19)	0.89 (0.65–1.21)	1.19 (0.74–1.93)	0.88 (0.51–1.52)	**0.62 (0.42–0.94)**	0.72 (0.46–1.13)
Middle	0.86 (0.68–1.10)	0.88 (0.66–1.16)	1.09 (0.66–1.80)	0.97 (0.55–1.69)	0.84 (0.57–1.25)	0.75 (0.49–1.17)
**Unstructured leisure (Ref = low level)**						
High	**3.24 (2.46–4.27)**	**3.89 (2.84–5.33)**	**7.65 (4.10–14.27)**	**5.5 (2.82–10.71)**	**7.00 (4.33–11.33)**	**7.11 (4.27–11.84)**
Middle	**2.65 (2.04–3.44)**	**2.67 (1.99–3.57)**	**3.52 (1.87–6.65)**	**2.76 (1.42–5.39)**	**2.93 (1.80–4.76)**	**2.70 (1.63–4.49)**
**Family leisure (Ref =low level)**						
High	**0.40 (0.32–0.51)**	**0.41 (0.31–0.53)**	**0.47 (0.28–0.79)**	**0.53 (0.30–0.95)**	**0.42 (0.29–0.62)**	**0.48 (0.32–0.73)**
Middle	**0.63 (0.51–0.79)**	**0.64 (0.49–0.82)**	0.74 (0.47–1.17)	0.69 (0.42–1.15)	**0.53 (0.35–0.81)**	**0.48 (0.30–0.76)**

^1^ Bold value show significant OR with *p* < 0.05.

**Table 6 ijerph-18-11477-t006:** Analysis of the maximum likelihood estimators of the model with simple effects and interactions between types of leisure.

Effect	YAU						YBD					
	The Netherlands		Peru		Spain		The Netherlands		Peru		Spain	
	Wald χ^2^	Pr > χ^2^	Wald χ^2^	Pr > χ^2^	Wald χ^2^	Pr > χ^2^	Wald χ^2^	Pr > χ^2^	Wald χ^2^	Pr > χ^2^	Wald χ^2^	Pr > χ^2^
Sex	4.34	0.04 *	2.84	0.09	2.34	0.13	14.10	<0.01 *	0.08	0.78	0.55	0.46
Family structure	5.78	0.02 *	4.45	0.04 *	6.88	0.01 *	5.34	0.02 *	4.71	0.03 *	1.29	0.26
Socioeconomical status	2.90	0.09	19.78	<0.01 *	0.61	0.43	7.64	0.01 *	7.72	0.01 *	0.43	0.51
Age group	322.63	<0.01 *	104.04	<0.01 *	200.24	<0.01 *	281.10	<0.01 *	55.53	<0.01 *	70.21	<0.01 *
*Self-control (SC)*	24.17	<0.01 *	11.81	<0.01 *	9.50	0.01 *	5.47	0.07	7.14	0.03 *	13.07	<0.01 *
Structured leisure (SL)	0.59	0.74	4.00	0.14	8.69	0.01 *	0.55	0.76	0.28	0.87	1.58	0.45
Unstructured leisure (UL)	65.37	<0.01 *	58.18	<0.01 *	51.09	<0.01 *	67.71	<0.01 *	22.81	<0.01 *	54.60	<0.01 *
Familiar leisure (FL)	46.01	<0.01 *	14.86	<0.01 *	21.51	<0.01 *	40.36	<0.01 *	1.94	0.38	12.56	<0.01 *
*Interaction SC × SL*	4.37	0.36	1.10	0.89	6.62	0.16	1.97	0.74	1.63	0.80	2.00	0.74
*Interaction SC × UL*	10.27	0.04 *	4.29	0.37	5.78	0.22	6.57	0.16	5.08	0.28	3.83	0.43
*Interaction SC × FL*	13.75	0.01 *	5.16	0.27	7.97	0.09	5.87	0.21	1.64	0.80	3.85	0.43

* *p* < 0.05.

## Data Availability

The data presented in this study are available on request from the corresponding author (Your Life Project) or the second author (LEF program). The data are not publicly available as they are still being exploited.

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
