# Peer review of "Adolescents’ Alcohol Use: Does the Type of Leisure Activity Matter? A Cross-National Study"

_ijerph, 2021, doi:10.3390/ijerph182111477_

Round 1
Reviewer 1 Report
In the manuscript “Adolescents’ alcohol use; does the type of leisure activity matter? A cross-national study”, the authors assessed the relationship between leisure activities, self-control and alcohol consumptions in adolescents from three countries. The results showed that unstructured leisure activities and family activities respectively increase and decrease the risk of alcohol consumptions and binge drinking in adolescents in all three countries; and level of self-control moderates the relationships in adolescents from the Netherlands. Some comments need to be addressed.
- For the last paragraph of Section 2.3, please add “sex” to the confounders.
- In Table 1, socioeconomic status was categorized into 3 levels - high, average, and low. However, only two levels (high, low) were used in the analysis. Please explain how scales were transformed in the method.
- Please describe how was self-control measured in the method.
- Section 3.1 does not read well. Please re-write the paragraph.
- Section 3.1, the author mentioned that more than half of the participants are boys, while in Table 2 and 3, the majority are girls. Please address this discrepancy and make sure the subjects were correctly coded. Considering this discrepancy, please revisit the results and confirm the OR is girl relative to boy.
- Line 306, since the test results are not significant, the estimated ORs should not be emphasized.
- Table 5, all numbers are bold, please fix it.
- Table 6, the variable “Etarian Group” is included, please add an introduction about this variable in the method.
- In the model including self-control, why is age excluded?
- The authors stated that people with a low level of structured leisure benefit from medium and high levels of self-control, and those with low levels of self-control benefited from increased family leisure. Are these conclusions reached through statistical tests or simply looking at the figures? Statistical results should be reported in parallel with the conclusion.
- Figure 1, please format it in the same way as figure 2, i.e., use UL as the vertical axis and self-control as the symbols/legend.
- Figure 1 and 2, please note it in captions that the figures are for the Netherlands only.
- Table 1, the abbreviation for Yearly binge drinking is missing.
- Line 231, BYD should be YBD.
- Please unify the use of YAC and YAU across the manuscript.
- Please unify the use of male/female, boy/girl, man/woman across the manuscript.
Author Response
Response to Reviewer 1 Comments
Point 1. For the last paragraph of Section 2.3, please add “sex” to the confounders.
Response 1. Thanks, added.
“sex was dichotomized in girls (0) and boys (1)”
Point 2. In Table 1, socioeconomic status was categorized into 3 levels - high, average, and low. However, only two levels (high, low) were used in the analysis. Please explain how scales were transformed in the method.
Response 2. Thanks for the observation. The low level was highly underrepresented in the sample, especially for Spain which was 2.4% or in the Netherlands which reached only 5.4%. Therefore, it was decided to merge this level with the average and for this variable only consider two values: low and high socioeconomic level. This explanation is incorporated into the text.
“However, the low socioeconomic level was highly underrepresented in the sample, especially for Spain which was 2.4% or in the Netherlands which reached only 5.4%. Therefore, it was decided to merge this level with the medium and for this variable only consider two values: low and high socioeconomic level”.
Point 3. Please describe how was self-control measured in the method.
Response 3. Thanks, added.
“Self-Control (SC) was measured through the evaluation of three behaviors (see Table 19), using five-point Likert-type scales, for the Netherlands: the response options were: 1 = Never, 2 = Rarely, 3 = Occasionally, 4 = Often, 5 = Very of-ten. For Spain and Peru the response options were: 1 = Never, 2 = Almost never, 3 = Sometimes, 4 = Almost always, 5 = Always. The responses were averaged, and these values were assigned to ordered tertiles (three groups of approximately 33% of individuals), indicating low, middle, and high average frequency of SC behaviors.”
Point 4. Section 3.1 does not read well. Please re-write the paragraph.
Response 4. Thanks for the observation. The paragraph was rewritten.
“The main characteristics of the participants are presented in Table 2. More than half of the participants are girls (59%). In the Netherlands, 57.9%, of the participants were between 15 and 17 years old while in the other two countries the majority were between 12 and 14 years old, in Peru 58.5% and Spain 55.2%. In all three countries, more than 80% of the young people involved had both parents and more than 60% had low socio-economic status. Peruvian and Spanish adolescents show a higher level of self-control (34.8% and 35.7% respectively) compared to Dutch (28.6%).”
Point 5. Section 3.1, the author mentioned that more than half of the participants are boys, while in Table 2 and 3, the majority are girls. Please address this discrepancy and make sure the subjects were correctly coded. Considering this discrepancy, please revisit the results and confirm the OR is girl relative to boy.
Response 5. Thanks for the observation. It is a mistake; we have changed girls to boys. It was corrected in the text.
Point 6. Line 306, since the test results are not significant, the estimated ORs should not be emphasized.
Response 6. Thanks for the observation. We agree, it is useless to make a discussion about non-significant parameters. However, the consistency of OR's less than unity, in all countries, we believed deserved a brief mention. We change this paragraph and remove any inference about the result.
“An interesting result is that the estimates of the ORs associated with structured leisure were less the unity, although without reaching statistically significant values.”
Point 7. Table 5, all numbers are bold, please fix it.
Response 7. Thanks for the observation. The format of the numbers has been corrected to show only the significant ORs in bold.
Point 8. Table 6, the variable “Etarian Group” is included, please add an introduction about this variable in the method.
Response 8. Thank you, it was a lack of coherence in the naming of the variables, “age group” is the same variable called “age group”. Names corrected.
Point 9. In the model including self-control, why is age excluded?
Response 9. Thank you for this observation. As it made us realize, in the previous observation, the variable "etarian group" actually refers to the variable "age group". So, age was included in the model with self-control. Naming was corrected.
Point 10. The authors stated that people with a low level of structured leisure benefit from medium and high levels of self-control, and those with low levels of self-control benefited from increased family leisure. Are these conclusions reached through statistical tests or simply looking at the figures? Statistical results should be reported in parallel with the conclusion.
Response 10. Thank you. Yes, this is the interpretation of the significant effect of the interaction. In the text it is already clarified with Wald statistic and its respective p-value.
“In the case of unstructured leisure (Fig 1.), people with a low level of unstructured lei-sure benefit from medium and high levels of self-control (w=10.27; p=0.04). In other words, if the level of self-control is low, the probability of alcohol consumption in-creases despite not being exposed to medium or high levels of unstructured leisure. Regarding the interaction between family leisure and self-control (Fig 2), it is noted how children with low levels of self-control especially benefited from increases in the levels of family leisure - even at the highest levels of family leisure - since this condition manages to neutralize the negative effects of the lack of SC (w=13.75; p=0.01) (Table 6).”
Point 11. Figure 1, please format it in the same way as figure 2, i.e., use UL as the vertical axis and self-control as the symbols/legend.
Response 11. Initially, the figures had the same format, however, when explaining the interaction, it was necessary to adjust them in order to make this effect evident. We would ask to keep the formats to facilitate the analysis of this effect.
Point 12. Figure 1 and 2, please note it in captions that the figures are for the Netherlands only.
Response 12. Thanks for the observation. The identification was added.
“Figure 1. Estimated probabilities for the interaction between self-control and unstructured leisure for the YAC in the Netherlands.”
“Figure 2. Estimated probabilities for the interaction between self-control and family leisure for the YAC in the Netherlands.”
Point 13. Table 1, the abbreviation for Yearly binge drinking is missing.
Response 13. Thanks for the observation. When doing the review, we believe that you are referring to table 5, which is where the abbreviation was missing. It was added.
“Table 5. Odds ratio and 95% confidence limits for Yearly Binge Drinking (YBD) in each country, raw and adjusted by confounders.”
Point 14. Line 231, BYD should be YBD.
Response 14. Thanks for the observation. It was a mistake, it was corrected.
“Initially, data was evaluated through descriptive statistics and then an association test for YAU and YBD was performed, with all the variables by a c2 test.”
Point 15. Please unify the use of YAC and YAU across the manuscript.
Response 15. Thank you for the observation. YAC and YAU were unified to YAPoint.
Point 16. Please unify the use of male/female, boy/girl, man/woman across the manuscript.
Response 16. Thank you for the observation. The alternatives of sex responses were unified to male/female.
Reviewer 2 Report
It was a very interesting and impressive study. This problem of Alcohol use and abuse has been analysed in a scientific manner. However, one of the important issues of paying for Alcohol could have been added. The Alcohol use is an expensive proposition and the costs cannot be borne by the teenagers alone. Good research paper!
Author Response
Response to Reviewer 1 Comments
Point 1. It was a very interesting and impressive study. This problem of Alcohol use and abuse has been analyzed in a scientific manner. However, one of the important issues of paying for Alcohol could have been added. The Alcohol use is an expensive proposition and the costs cannot be borne by the teenagers alone. Good research paper!
Response 1. Thank you for your kind words. This is indeed an important issue to mention in the paper. We have added this in the Discussion:
‘In addition, access to alcohol may also be related to family leisure activities as a protective and unstructured leisure as a risk factor.’
Reviewer 3 Report
Review – ijerph-1377170
Adolescents’ alcohol use; Does the type of leisure activity matter? A cross-national study
GENERAL COMMENT:
The present manuscript assessed the associations of various types of leisure activities with two indicators of adolescents’ alcohol consumption and, which is definitely a unique feature, investigated possible moderation effect of self-control. The authors possessed quite an impressive sample of adolescents from three different countries. Moreover, the topic itself is of interest to readers interested in adolescents’ time use, substance use, as well as development in general. However, the manuscript is full of various inconsistencies and signs of lack of attention to details. The Introduction provides numerous important arguments for studying the topic but it is quite vague in some passages. The Methods section lacks multiple crucial pieces of information (e.g. source or psychometric properties of the measures used). In particular, I am doubtful about the analytic design (or more precisely, way of treating the variables) and the way of ‘merging data files from distinct countries’ with different procedures of sampling, as well questionnaires used. This makes me question the informative value of large part of the Results. For these reasons, I cannot recommend the manuscript for publication. Please refer to my comments below (please note that I selected only some examples and paid much less attention to review from the Results section onwards):
SPECIFIC COMMENTS:
- Introduction (p. 2; l. 44) – ‘start drinking alcohol before the age of 15.’ I suggest toning the phrase down. The present wording implies that the adolescents actually adopt a habit of regular alcohol consumption at this age and not that they usually only acquire their first experience in this early age (I admit that it is risky anyway).
- Introduction (p. 2; l. 50) – Please add age of the Dutch and Spanish ‘binge-drinkers’. The reported prevalence appears too high… even in the light of the authors’ own results.
- Introduction (p. 2; l. 55–57) – I wondered why the aim of the study is presented already here + why it does not contain family leisure. I’d suggest deleting it here and leave it only in the last paragraph of the Introduction.
- Introduction (p. 2; l. 60–62) – I find this sentence very misleading. There were quite a few studies investigating the associations of involvement in various leisure activities and alcohol consumption – mostly from the U.S. (for instance, check somewhat older review from 2012 –doi:10.1016/j.dr.2011.10.001), but also from Europe (e.g. doi:10.1007/s00038-018-1125-3; doi:10.1111/1745-9125.12105) or Japan (e.g. doi:10.1007/s00038-015-0697-4).
- Introduction (p. 2; l. 65) – Which types of SL? Please be more specific. As far as I know, SL is generally considered beneficial for health (see e.g. doi:10.1136/jech.2009.08 or doi: 10. 1136/jech- 2020- 215319), not to speak about positive youth development (tons of the U.S. literature).
- Introduction (p. 2; l. 77–83) – Please be more specific about UL activities. You generally label them as risky but this does not apply to all such activities. It concerns especially these that are ‘peer-oriented’ and do not have ‘skill-building aims’, e.g. hanging out mentioned on p.2; l. 84. Please see e.g. doi:10.1080/17430437.2018.1472242. In addition, as far as I am familiar with these two references (27 & 28), they do not provide evidence for your statement and rather advocate for participation in UL activities.
- Introduction (p. 2; l. 94) – What individual characteristics do you mean?
- Procedure (p. 3; l. 128) – When was the data collected? How many schools were there in total in Spain and Peru? What was the response rate both at the level of schools and individuals? The latter should be ideally stratified by country + reported also for the Netherlands.
- Procedure (p. 4; l. 158) – Really only two schools in the Netherlands? There were over 2000 respondents from only two schools?
- Participants (p. 4; l. 171) – ‘Convenience sampling’… this contradicts the information that all school were invited (p. 3; l. 125).
- Participants (p. 4; l. 184) – The sentence is unfinished. Moreover, I wondered why it is presented in text for only one country. Consistency helps readers navigate in the text.
- Measures (general) – As mentioned in the general comment, I miss information on sources of measures used in the present study and, ideally, also their validity and reliability. For instance, family leisure activities resemble a list used originally by Prof. Helen Sweeting but it is not referred to anywhere.
- Measures (p. 4; l. 190–193) – Given the response options, it does not really seem to measure yearly consumption but rather monthly. Moreover, the response option ‘1=at least one day a month’ has a different wording in Table 1: ‘less than one day a month’, which is quite a difference. Btw. have you tested the sensitivity using the ‘cut-off point of repeated YAC’, i.e. more than just once because experimenting with alcohol is quite normative and a single occasion does not necessarily indicate risk behaviour.
- Measures (p. 4; l. 198–199) – How exactly was exercising investigated in the questionnaire. It certainly does not have to be SL (in terms of the definition in the Introduction, i.e. adult-supervised, schedule…). The same applies to cultural/artistic activities.
- Measures (p. 5; l. 202–203) – I see the point of creating tertiles (this is an actual correct spelling) in order to compare data from different countries. However, I find it very questionable. Just for instance, if a respondent indicated ‘3=one day a week’ for all three activity types, the sum score is 9, whereas somebody heavily involved in just a single activity (5=every day) but not at all in other activities (1=never) has the sum score of 7, i.e. lower level of SL participation, which is actually not true. This applies to UL, as well as family activities. Moreover, it seemed to me (based on Table 2) that the tertiles were done for a total sample, which is an unfortunate decision (and I think if you need to do that, you should do so after stratification by country). Anyway, I strongly recommend to reconsider the categorisation of all the leisure-related variables (e.g. number of those done at least multiple times a week).
- Measures (p. 5; l. 223) – How do schools assess the status of families? Who provided that information? There is no mention about any questionnaire for school administrators.
- Table 1 – The first two columns are ‘truncated’, i.e. some words are missing in multiple rows.
- Measures (Table 1) – It is not explained how was self-control measured. What were the response options?
- Analysis – Given that alcohol consumption is strongly influenced by peers, did you take the effect of school/class into account in the regression analyses (i.e. multilevel analysis)?
- Results (p. 7; l. 245) – Table 2 shows that there are more girls in the sample.
- Results (p. 7; l. 246–247) – Majority of Dutch between 15 and 17, whereas Mean age reported above was 14.7 appears unlikely. Moreover, given that in Peru/Spain more than half of sample were 12–14, I wondered why the median age value was selected. Last but not least, taking this cross-national into account, higher alcohol consumption observed in the Netherlands is not surprising, is it?
- Results (p. 7; l. 247) – More than 60% low SES? Is this reflective of the population in any of the countries involved in the study?
- Results (general) – As you mention in the limitations, you possessed cross-sectional data. For this reason, please avoid using ‘causal language’ (effect, protective, etc.).
- Results (p. 10; l. 306) – the OR in Spain was not statistically significant in Table 5.
- Results (p. 10; l. 307) – ‘…protective in all countries…’ This is disputable in Peru and, in fact, in the Netherlands too.
- Discussion (p. 386–395) – I would certainly include difference between questionnaires across countries as a limitation, at least (if you insist on combining the samples). Moreover, I strongly doubt that you can say your sample was representative (2 schools in the Netherlands, convenience sampling in Peru, no information on response rates…)
MINOR/DISCRETIONARY COMMENTS:
- Title – Replace semicolon by colon.
- Introduction (p. 2; l. 49) – Please add ‘… within a limited period of time”. In addition, quarter should be plural, i.e. ‘quarters’.
- Introduction (p. 2; l. 56) – Please write in full ‘structured & unstructured’.
- Introduction (p. 2; l. 97) – Either add a verb to the sentence or replace ‘amongst which’ by ‘including’.
- Procedure (p. 3; l. 137–138) – ‘Anonymous… did not provide names or…’ This sentence seems tautological to me.
- Participants (p. 4; l. 174) – The link does not work. The following ones seems OK: http://proyectoyourlife.com/en/
- Measures (p. 4; l. 192–193) – Response option number 13 is missing in the brackets.
- Materials and Methods & Results – There are several instances when the YAU and YBD are misspelled.
- Analysis (p. 7; l. 241) – ‘alpha’ instead of ‘P value’.
- Results (p. 7; l. 248) – ‘level’ instead of ‘novel’
- Results (p. 9; l. 278) – ‘comparison’ instead of ‘relation’.
- Table 5 – I could not identify which values were bold.
- Results (p. 10; l. 308–309) – Such an explanatory sentence should not be in the Results.
Author Response
Response to Reviewer 3 Comments
Point 1. Introduction (p. 2; l. 44) – ‘start drinking alcohol before the age of 15.’ I suggest toning the phrase down. The present wording implies that the adolescents actually adopt a habit of regular alcohol consumption at this age and not that they usually only acquire their first experience in this early age (I admit that it is risky anyway).
Response 1. This is indeed true. We added ‘experimenting with’ to the sentence.
‘On average, 58% of European and Canadian adolescents start experimenting with drinking alcohol before the age of 15.’
Point 2. Introduction (p. 2; l. 50) – Please add age of the Dutch and Spanish ‘binge-drinkers’. The reported prevalence appears too high… even in the light of the authors’ own results.
Response 2. The prevalence of binge-drinkers indeed seems high, yet this is the prevalence of binge-drinkers among adolescents who have drunk alcohol in the previous month, thus regular drinkers. We have added the ages of the groups binge-drinkers.
‘Nearly three quarter (70.8%) of the Dutch youth <16 years [6] and more than half (56%) of the youth <15 years Spanish adolescents [7] who had drunk alcohol in the previous month, had been involved in binge-drinking’.
Point 3. Introduction (p. 2; l. 55–57) – I wondered why the aim of the study is presented already here + why it does not contain family leisure. I’d suggest deleting it here and leave it only in the last paragraph of the Introduction.
Response 3. We prefer to make the aim of the study clear right from the beginning of the manuscript. Yet, as preferred by the reviewer, we have deleted this here and only describe the aim of the study at the end of the Introduction.
Point 4. Introduction (p. 2; l. 60–62) – I find this sentence very misleading. There were quite a few studies investigating the associations of involvement in various leisure activities and alcohol consumption – mostly from the U.S. (for instance, check somewhat older review from 2012 –doi:10.1016/j.dr.2011.10.001), but also from Europe (e.g. doi:10.1007/s00038-018-1125-3; doi:10.1111/1745-9125.12105) or Japan (e.g. doi:10.1007/s00038-015-0697-4).
Response 4. There is indeed much more research available. Yet, these studies have not included these three different types of leisure activities; the independent role of each type of activity was related to adolescents’ drinking. We have now described this more explicitly.
‘Although not much research is available on the relative relations between specific types of leisure activity and use of alcohol, there is research conducted that investigated the role of unstructured leisure activities [15], [16]’
Point 5. Introduction (p. 2; l. 65) – Which types of SL? Please be more specific. As far as I know, SL is generally considered beneficial for health (see e.g. doi:10.1136/jech.2009.08 or doi: 10. 1136/jech- 2020- 215319), not to speak about positive youth development (tons of the U.S. literature).
Response 5. This is true indeed. We have reformulated this sentence. It now reads:
‘Overall, SL is considered beneficial for health and support a positive youth development [19–21].’
Point 6. Introduction (p. 2; l. 77–83) – Please be more specific about UL activities. You generally label them as risky but this does not apply to all such activities. It concerns especially these that are ‘peer-oriented’ and do not have ‘skill-building aims’, e.g. hanging out mentioned on p.2; l. 84. Please see e.g. doi:10.1080/17430437.2018.1472242. In addition, as far as I am familiar with these two references (27 & 28), they do not provide evidence for your statement and rather advocate for participation in UL activities.
Response 6. We appreciate the comment, as the reviewer indicates we discuss this at the end of the paragraph as a matter of order, we move from the general approach to the specifics.
Point 7. Introduction (p. 2; l. 94) – What individual characteristics do you mean?
Response 7. Thank you for the observation. We are referring to Self-Control. It was added.
“The differential susceptibility hypothesis stipulates that some adolescents are more susceptible to both positive and negative environmental effects than others, depending on, for example, individual characteristics [40], such as adolescents’ self-control.”
Point 8. Procedure (p. 3; l. 128) – When was the data collected? How many schools were there in total in Spain and Peru? What was the response rate both at the level of schools and individuals? The latter should be ideally stratified by country + reported also for the Netherlands.
Response 8. This information it was added.
“Spain and Peru. During the period between December 2016 and September 2018, a convenience sampling was carried out in schools in those countries, since it is an on-going dynamic and multi-purpose international study. Schools were invited by e-mail, 18 for Spain and 15 for Peru, giving them the link to the website, which was designed to provide detailed information to the participants (https://proyectoyourlife.com//). The surveys were completed in the classroom, under the supervision of the teacher in charge, since the evaluation instrument considers, in all its stages, the child's free decision to participate or not, it is considered that the response rate was 100%.”
“The Netherlands. The Dutch data were collected in May 2018. 2166 adolescents filled out the questionnaire (96% response rate).”
Point 9. Procedure (p. 4; l. 158) – Really only two schools in the Netherlands? There were over 2000 respondents from only two schools?
Response 9. Yes, two large schools were included. In the Netherlands, schools that offer all levels of education often have more than 1000 students.
Point 10. Participants (p. 4; l. 171) – ‘Convenience sampling’… this contradicts the information that all school were invited (p. 3; l. 125).
Response 10. Thanks for the observation. To avoid confusion in the use of this term, we refer the sampling to the non-probability group. With this we are clear that biases are possible, which is part of the limitations of this study.
‘An invitation with information about the project was sent by e-mail to public and private high schools. In Spain, invitations were sent to all high schools available in public databases. In Peru, local collaborators include schools with easier access. In this way, the sampling presented is non-probabilistic.’
Point 11. Participants (p. 4; l. 184) – The sentence is unfinished. Moreover, I wondered why it is presented in text for only one country. Consistency helps readers navigate in the text.
Response 11. There was a ‘dot’ missing at the end, we have added this now. Also, the percentage of boys was also added for the other countries in text.
Point 12. Measures (general) – As mentioned in the general comment, I miss information on sources of measures used in the present study and, ideally, also their validity and reliability. For instance, family leisure activities resemble a list used originally by Prof. Helen Sweeting but it is not referred to anywhere.
Response 12. Two references were added to the description of the leisure activity measures.
Point 13. Measures (p. 4; l. 190–193) – Given the response options, it does not really seem to measure yearly consumption but rather monthly. Moreover, the response option ‘1=at least one day a month’ has a different wording in Table 1: ‘less than one day a month’, which is quite a difference. Btw. have you tested the sensitivity using the ‘cut-off point of repeated YAC’, i.e. more than just once because experimenting with alcohol is quite normative and a single occasion does not necessarily indicate risk behaviour.
Response 13. Thanks for the comment. For the three countries, the question referred to use in the last year. "¿Prevalence alcohol use- Past 12 months?" for the Netherlands and "¿In the last 12 months, how often have you done the following activities?" for Spain and Peru. We add an extra clarifying note.
Regarding the cut-off point, the response variable was no consumption against some consumption. We consider that for a minor the mere report of alcohol consumption is already problematic.
‘For the scales of Yearly Alcohol Use (YAU) and Yearly Binge Drinking (YBD), participants were asked to report the number of drinking occasions where, for YAU, they consumed at least one alcoholic beverage during the last year, and, for YBD, occasions where they consume 5 or more alcoholic drinks in few hours in the last year [53].’
Point 14. Measures (p. 4; l. 198–199) – How exactly was exercising investigated in the questionnaire. It certainly does not have to be SL (in terms of the definition in the Introduction, i.e. adult-supervised, schedule…). The same applies to cultural/artistic activities.
Response 14. SL is characterized also by providing a certain structure aiming at skill building. That is the part that is applying to SL. In most cases, sport activities are also under supervision of an adult. This broader description can be found in the Measurements section.
‘Supervised Leisure (SL) indicates the number of activities that adolescents engage in under supervision of adults, those with a structure, and/or those with a certain structure, regular schedule, clearly defined goals and rules, and focused on building skills [18]. The instrument developed by Carlos et al. [46] was used to ask for engagement in a variety of activities; both structured and unstructured.’
Point 15. Measures (p. 5; l. 202–203) – I see the point of creating tertiles (this is an actual correct spelling) in order to compare data from different countries. However, I find it very questionable. Just for instance, if a respondent indicated ‘3=one day a week’ for all three activity types, the sum score is 9, whereas somebody heavily involved in just a single activity (5=every day) but not at all in other activities (1=never) has the sum score of 7, i.e. lower level of SL participation, which is actually not true. This applies to UL, as well as family activities. Moreover, it seemed to me (based on Table 2) that the tertiles were done for a total sample, which is an unfortunate decision (and I think if you need to do that, you should do so after stratification by country). Anyway, I strongly recommend to reconsider the categorisation of all the leisure-related variables (e.g. number of those done at least multiple times a week).
Response 15. Yes, in fact another statistic could have been used, such as the maximum value, as suggested in the comment. However, to clarify, in this case the mean value was used instead of the sum of the scores, which although it can punish specific behaviors such as those mentioned, it allows a weighting of a great variety of activities. Therefore, in that sense the indicator would favor diversity in leisure activities rather than concentration on a single activity.
Moreover, these average values were ranked separately by country, so that for each country there are young people who, on average, responded with high, medium and low average frequency in leisure activities.
It should be mentioned that the results obtained allow the measurement system to be validated, since it was sensitive to all the research hypotheses raised. However, it can be considered as a suggestion for future work to evaluate other types of statistics (maximum, minimum, etc.).
Point 16. Measures (p. 5; l. 223) – How do schools assess the status of families? Who provided that information? There is no mention about any questionnaire for school administrators.
Response 16. Adolescents reported about the family status themselves; it is referring to the family situation. This is also the case for level of education; this is also adolescent reported.
Point 17. Table 1 – The first two columns are ‘truncated’, i.e. some words are missing in multiple rows.
Response 17. Thanks for the observation. It was a table setting problem in Word. It is solved by increasing the height of the rows.
Point 18. Measures (Table 1) – It is not explained how was self-control measured. What were the response options?
Response 18. Thanks for the observation, added.
“Self-Control (SC) was measured through the evaluation of three behaviors (see Table 19), using five-point Likert-type scales, for the Netherlands: the response options were: 1 = Never, 2 = Rarely, 3 = Occasionally, 4 = Often, 5 = Very of-ten. For Spain and Peru the response options were: 1 = Never, 2 = Almost never, 3 = Sometimes, 4 = Almost always, 5 = Always. The responses were averaged, and these values were assigned to ordered tertiles (three groups of approximately 33% of individuals), indicating low, middle, and high frequency of SC behaviors.”
Point 19. Analysis – Given that alcohol consumption is strongly influenced by peers, did you take the effect of school/class into account in the regression analyses (i.e. multilevel analysis)?
Response 19. Yes, it is a point that can be improved, but in this case the analysis was evaluated with the assumption of independence. The proposed model of logistic regression with interactions was already complex enough and to keep the parsimony of the model to a minimum we chose not to consider these nested effects. However, knowing that the correlations between the observations could increase the confidence intervals (peers’ effect), we performed the Bootstrap procedure that would improve the estimation of the parameters without increasing the number of them.
Point 20. Results (p. 7; l. 245) – Table 2 shows that there are more girls in the sample.
Response 20. There is a mistake, we have changed girls to boys
Point 21. Results (p. 7; l. 246–247) – Majority of Dutch between 15 and 17, whereas Mean age reported above was 14.7 appears unlikely. Moreover, given that in Peru/Spain more than half of sample were 12–14, I wondered why the median age value was selected. Last but not least, taking this cross-national into account, higher alcohol consumption observed in the Netherlands is not surprising, is it?
Response 21. Thanks for the observation. Table 2 shows that the separation of the groups according to the median achieved a relatively homogeneous distribution in the three countries. This was the criterion that allowed us to validate the use of the median as a value to separate the age groups. And, indeed, it is not surprising that the drinking rates are higher in the Netherlands.
Point 22. Results (p. 7; l. 247) – More than 60% low SES? Is this reflective of the population in any of the countries involved in the study?
Response 22. Thanks for the observation. Originally the low level was highly underrepresented in the sample, especially for Spain which was 2.4% or in the Netherlands which reached only 5.4%. Therefore, it was decided to merge this level with the average and for this variable only consider two values: low and high socioeconomic level. This explanation is incorporated into the text.
“However, the low socioeconomic level was highly underrepresented in the sample, especially for Spain which was 2.4% or in the Netherlands which reached only 5.4%. Therefore, it was decided to merge this level with the medium and for this variable only consider two values: low and high socioeconomic level”.
Point 23. Results (general) – As you mention in the limitations, you possessed cross-sectional data. For this reason, please avoid using ‘causal language’ (effect, protective, etc.).
Response 23. Only in the Results section we describe the findings in terms of ‘protective’ and ’effect’, in addition to ‘relations’ and ‘associations’, as this is the way it is tested. I was tested as the leisure activities effects alcohol use. Therefore, we mentioned this as a limitation in the Discussion section and avoided this terminology in the discussion of the results.
Point 24. Results (p. 10; l. 306) – the OR in Spain was not statistically significant in Table 5.
Response 24. Thanks for comment. It was corrected.
‘that the ORs by sex already show women with risk values in relation to men in the Netherlands (1.47).’
Point 25. Results (p. 10; l. 307) – ‘…protective in all countries…’ This is disputable in Peru and, in fact, in the Netherlands too.
Response 25. Thanks for the observation. We agree, it is useless to make a discussion about non-significant parameters. However, the consistency of OR's less than unity, in all countries, we believed deserved a brief mention. We change this paragraph and remove any inference about the result.
‘An interesting result is that the estimates of the ORs associated with structured leisure were less the unity, although without reaching statistically significant values.’
Point 26. Discussion (p. 386–395) – I would certainly include difference between questionnaires across countries as a limitation, at least (if you insist on combining the samples). Moreover, I strongly doubt that you can say your sample was representative (2 schools in the Netherlands, convenience sampling in Peru, no information on response rates…)
Response 26. We do agree that the included samples were not representative of the whole Dutch/Spanish/Peruvian population. We excluded the sentence about the representative sample.
MINOR/DISCRETIONARY COMMENTS:
Point 27. Title – Replace semicolon by colon.
Response 27. Thank you, corrected.
Point 28. Introduction (p. 2; l. 49) – Please add ‘… within a limited period of time”. In addition, quarter should be plural, i.e. ‘quarters’.
Response 28. Thank you, corrected.
Point 29. Introduction (p. 2; l. 56) – Please write in full ‘structured & unstructured’.
Response 29. We rewrite the paragraph.
‘It is important to investigate protective factors that may contribute to lowering drinking levels among youth [12], such as leisure activities.’
Point 30. Introduction (p. 2; l. 97) – Either add a verb to the sentence or replace ‘amongst which’ by ‘including’.
Response 30. Thanks, it was modified.
Point 31. Procedure (p. 3; l. 137–138) – ‘Anonymous… did not provide names or…’ This sentence seems tautological to me.
Response 31. Thanks, we modified the paragraph.
‘Adolescents filled out a self-report questionnaire in which participants did not provide names or any other personally identifiable information.’
Point 32. Participants (p. 4; l. 174) – The link does not work. The following ones seems OK: http://proyectoyourlife.com/en/
Response 32. Yes, thanks, it may be the Word configuration that when including the line change script, damaged the link. It is already corrected.
Point 33. Measures (p. 4; l. 192–193) – Response option number 13 is missing in the brackets.
Response 33. Thanks, corrected.
Point 34. Materials and Methods & Results – There are several instances when the YAU and YBD are misspelled.
Response 34. Thanks, corrected.
Point 35. Analysis (p. 7; l. 241) – ‘alpha’ instead of ‘P value’.
Response 35. Thanks, corrected.
Point 36. Results (p. 7; l. 248) – ‘level’ instead of ‘novel’
Response 36. Thanks, corrected.
Point 37. Results (p. 9; l. 278) – ‘comparison’ instead of ‘relation’.
Response 37. Thanks, corrected.
Point 38. Table 5 – I could not identify which values were bold.
Response 38. Thanks, corrected.
Point 39. Results (p. 10; l. 308–309) – Such an explanatory sentence should not be in the Results.
Response 39. Thanks, it was eliminated.
Round 2
Reviewer 1 Report
All of my comments have been addressed. No further questions.
Author Response
Dear reviewer:
Thanks for the feedback. They certainly helped improve the quality of the paper.
Sincerely,
on behalf of all authors,
Edgar Benitez
Reviewer 3 Report
Review – ijerph-1377170
Adolescents’ alcohol use; Does the type of leisure activity matter? A cross-national study
GENERAL COMMENT:
The authors addressed several my comments satisfactorily. However, quite a few issues were left unresolved or insufficiently explained from my point of view. Moreover, some new comments arise from the newly added text. All in all, I still cannot recommend the manuscript for publishing, despite the authors’ obvious efforts to improve it. The comments below are not exhaustive; I was unable to go through the entire manuscript thoroughly in such a short time.
“UNRESOLVED” COMMENTS FROM THE REVIEW:
- Introduction (p. 2; l. 84) – My former comment no. 6 on unsuitability of the two references (currently no. 30 & 31; previously 27 & 28) remained unresolved. These two studies actually rather support participation in unstructured leisure activities. In one instance, it’s about “positive effects” of skating and skate parks; in the second one, the authors generally showed “beneficial effects” of unstructured activities such as hiking, fishing, volunteering or arts (not associated with club), but hanging out as well.
- Introdcution (p. 2; l. 78–92) – I do not think that the authors captured what I meant in my original comment no. 6. My point was that unstructured leisure cannot be generally labelled as risky because this applies only to the activities that are, in general, not supervised and have a strong ‘peer-oriented’ socialising character. See the studies mentioned in the current comment no. 1 above.
- Participants (p. 4; l. 177–178) – The information on the response rate is completely unacceptable in my opinion. Firstly, the authors did not provide information on school level response rate – RR (as requested in my former comment no. 8). Secondly (which is far more important), the authors mention 100% response rate in Spain and Peru. I’ve taken it always for granted that the option not to participate is (or at least should be) integral to any survey conducted. However, this does not establish 100% RR. If anybody declines to participate, RR automatically decreases. I’ve never personally encountered or read about any slightly larger study that would achieve 100% RR. Given that the schools were apparently the primary sampling unit, RR should stand for number of completed questionnaires (“divided by”) number of students registered in the schools (or classes involved in the survey), thus reflecting those who declined to participate or were missing, for example, due to illness.
- Participants (p. 4; l. 187) – My previous comment no. 11 about why rate of boys was presented only for the Dutch remained unresolved, despite authors claiming it was added to the text (at least I was unable to find it).
- Measures (p. 5; 214+) – The authors did not reflect my original comment no. 15 on treatment of leisure-related variables (structured, unstructured, and family) in the manuscript (i.e. use another statistical apparoach) and I find their explanation, as well as the way they resolved it in the manuscript, unsatisfactory. Given that the response options of all the leisure-related variables is ordinal, it makes no sense to compute an average here. Concerning their response to my comment (diversity of activities), it is very misleading to call it an ‘average frequency’ anyway because what they apparently intended to capture would rather correspond to so call breadth of participation.
- Measures (p. 5; l. 231–237) – I appreciate that the authors added the description of the “Self-control” measure. However, there is no information on source of the question or its psychometric properties or its previous use (requested previously in the comment no. 12).
- Results (p. 8; l. 278–279) Previously I commented (no. 21) that it is not OK to stress that the Dutch showed higher rates of YAU and YBD because of higher age but it remained here without controlling for the factor of age.
- Discussion (p. 13; l. 412+) – The authors removed the sentence about representativeness of their data but did not consider the rest of the comment (differences in questionnaire, regardless of language/semantic standardisation) or reflect the “cross-sectional” nature of data, thus using the words “risk/protective” factor even here.
NEW COMMENTS:
- Introduction (p. 2; l. 62) – As I was one of the authors of this paper (ref. no. 15), I’m sure that it is an irrelevant reference in this context. We focused on organized/structured activities and well-being and not on unstructured leisure and alcohol.
- Participants (p. 4; l. 184) – I do not see the point of excluding the information on two schools from the Edam-Volendam municipality (regardless of my original comment no. 9). Actually, I think it needs to be provided to the readers because it is crucial to make an objective impression of the Dutch sample.
- Measures (p. 4; l. 192–193) – As a member of the HBSC network, I have several questions regarding this new text. Why didn’t you use standardised and revised (i.e. officially approved) translations of the questions. There is a rigorous process of back-translation in place in the HBSC network, which ensures/facilitates cross-national comparability of the survey. What was the point of adapting the language version later to English? The original (i.e. source) version of all the questions included in the HBSC is English. Most crucial, I could not identify any of the HBSC questions in the Measures used in the reviewed study. The “closest” one was the YAU in the Netherlands. However, in the 2009/10 (as cited ref. no. 51) there were only 7 response options (never; 1–2 times, 3–5 times; 6–9 times; 10–19 times, 20–39 times; 40 or more times) for this particular question. Lastly, this question has not been used since 2010 and another one is in use since 2013/14 (the wording, as well as the response options have changed substantially)... Taken the information in this comment into account, I do not find it appropriate to refer to the HBSC study at all.
- Terminology – The authors use girls/boys here, whereas male/female distinction is present in the Tables. In the same sense, it appears to me that structured/unstructured vs. supervised/unsupervised are used interchangeably in the text, while it is not clear to me whether these are supposed to mean the same or not.
- Measures (p. 5; l. 206) – I’m sorry. I forgot to include this comment in my original review. Why is structured mentioned twice here?
- Measures (p. 5; l. 212–215) – In my former comment no. 15, I asked if the tertiles were created for a total sample or split by countries. Such a piece of information should be explicitly stated in the manuscript to facilitate its comprehension.
- Measures (p. 5; l. 241–242) – Although the authors indicate that the socioeconomic status was self-reported, the manuscript still mentions that this was the case only in the Netherlands and not in Spain and Peru.
- Results (p. 7; l. 271) – The authors newly recognize that low socioeconomic status was significantly underrepresented in the sample (In the Methods section) but still keep labelling it as low in the Results.
- Results (p. 10; l. 329–331) – I do not understand the sentence. What does “less the unity” mean?
- Discussion (p. 13; l. 373) – Despite claiming that the authors avoided use of “causal language” in the Discussion, it repeatedly occurs there, with the first occurrence at the end of the first paragraph.
- References – There is unnecessary spacing between lines + I was unable to find ref no. 56 in the text + the link for ref no. 51 was not working on my computer.
Author Response
Dear reviewer:
Thanks for the feedback. They certainly helped improve the quality of the paper.
Sincerely,
on behalf of all authors,
Response to Reviewer 3 Comments,
2nd Round
We thank the reviewer again for his/her helpful comments and suggestions to further improve the manuscript. We hope we have been able to address them in such a way that the reviewer is satisfied.
Point 1. Introduction (p. 2; l. 84) – My former comment no. 6 on unsuitability of the two references (currently no. 30 & 31; previously 27 & 28) remained unresolved. These two studies actually rather support participation in unstructured leisure activities. In one instance, it’s about “positive effects” of skating and skate parks; in the second one, the authors generally showed “beneficial effects” of unstructured activities such as hiking, fishing, volunteering or arts (not associated with club), but hanging out as well.
Response 1. Following the reviewer's indication, which we inadvertently did not reply to, we have replaced the reference to these studies and now reads:
“Participation in UL can be considered positive in certain unsupervised activities, with a strong socialising character among peers [30], [31]. On the other side, according to numerous studies, time spent "hanging out" and lacking of participation in organized activities predicts delinquency”.
Point 2. Introduction (p. 2; l. 78–92) – I do not think that the authors captured what I meant in my original comment no. 6. My point was that unstructured leisure cannot be generally labelled as risky because this applies only to the activities that are, in general, not supervised and have a strong ‘peer-oriented’ socialising character. See the studies mentioned in the current comment no. 1 above.
Response 2. Thank you for your comment. We consider that we have dealt with it in the previous section.
Point 3. Participants (p. 4; l. 177–178) – The information on the response rate is completely unacceptable in my opinion. Firstly, the authors did not provide information on school level response rate – RR (as requested in my former comment no. 8). Secondly (which is far more important), the authors mention 100% response rate in Spain and Peru. I’ve taken it always for granted that the option not to participate is (or at least should be) integral to any survey conducted. However, this does not establish 100% RR. If anybody declines to participate, RR automatically decreases. I’ve never personally encountered or read about any slightly larger study that would achieve 100% RR. Given that the schools were apparently the primary sampling unit, RR should stand for number of completed questionnaires (“divided by”) number of students registered in the schools (or classes involved in the survey), thus reflecting those who declined to participate or were missing, for example, due to illness.
Response 3. Concerning the response rate in our study we would like to make the following considerations:
Our previous response stating a 100% response rate was made considering that all kids that participated in our survey either responded to our questions or responded "I do not want to respond". However, we agree that, in reality, we can never know what the real denominator is in a convenience sample and thus it does not really make sense to give a response rate because we do not have an initial sampling frame (or the frame is infinite).
The response rate refers to the number of persons who complete a survey divided by the number of people who make up the total sample group. This is especially relevant in studies where selection bias can be a problem and is the reason why this information is requested and given in such studies (for example clinical trials or representative studies that could be using a probability sampling technique).
The Yourlife project is not a representative survey and is more like a convenience sample or a cohort study where response rates (as defined above) are not relevant. In such studies representativeness is not a goal and thus the response rate is not an issue. In these studies, follow up rates are the issue. The analysis we are presenting does not entail follow up data. In cohort and convenience samples the presence of confounding is dealt with using multivariate adjustments and thus having large sample sizes is more important. We have achieved this goal.
The Netherlands data can give an estimate of response rate because the initial target sample were students in two high schools. This is a closed initial sampling frame. But in their case as well it is more important to be able to adjust for confounding as representativeness is not a goal either.
A convenience sample is a type of non-probability sampling method where the sample is taken from a group of people easy to contact or to reach. The researcher puts out a request for members of a population to join the sample, and people decide whether or not to be in the sample. In our case, we approach schools and allow them to participate (whoever wants to participate does so). There is no "total sampling group" as such because this is an open-end study where the sample sizes increase as schools participate. The goal of such studies is simply to obtain large samples and to be able to adjust for confounding to consider any selection bias in the participation process (we have schools from different socio economical strata, etc.).
We propose to add a summary of these comments in the discussion section in order for the reader to be better informed on how they should interpret our data.
We have added this text to the discussion:
“Some comments are worthy concerning the response rates in our analysis. The Your Life project is not a representative survey, being more like a convenience sample, where response rates are not relevant. In such studies, representativeness is not a goal and thus the response rate is not an issue, but follow up rates are. Since the analysis we are presenting does not entail follow up data, the presence of confounding factors is addressed by multivariate adjustment, so large sample sizes are more important, and this goal has been achieved in this study. The Netherlands data has an estimate of the response rate because the initial target sample consisted of students from two schools. This was thus a closed sampling frame. However, also in their case it was more important to adjust for confounding factors, as representativeness was not an objective either.”
Point 4. Participants (p. 4; l. 187) – My previous comment no. 11 about why rate of boys was presented only for the Dutch remained unresolved, despite authors claiming it was added to the text (at least I was unable to find it).
Response 4. Thank you for your suggestion. We have added the percentage of boys.
Point 5. Measures (p. 5; 214+) – The authors did not reflect my original comment no. 15 on treatment of leisure-related variables (structured, unstructured, and family) in the manuscript (i.e. use another statistical approach) and I find their explanation, as well as the way they resolved it in the manuscript, unsatisfactory. Given that the response options of all the leisure-related variables is ordinal, it makes no sense to compute an average here. Concerning their response to my comment (diversity of activities), it is very misleading to call it an ‘average frequency’ anyway because what they apparently intended to capture would rather correspond to so call breadth of participation.
Response 5. We appreciate your observation on the focus of the analysis, however, we consider that the approach we give has the necessary validity. The problem of working with ordinal variables is clear, in fact almost all psychometric studies are based on ordinal scales. However, the use of parametric models is generally accepted where the mean and variance are the basis for all analyses. As other authors have presented, this work validates the use of the mean in two assumptions: the central limit theorem (n = 4608) and that the skewness and kurtosis values were between the range of -2 to 2, as summarized by Lloret-Segura et. al., 2012, in his state of the art on this methodology.
Lloret-Segura, S., Ferreres-Traver, A., Hernández-Baeza, A., & Tomás-Marco, I. (2014). El análisis factorial exploratorio de los ítems: una guía práctica, revisada y actualizada. Anales de Psicología/Annals of Psychology, 30(3), 1151-1169.
Regarding the term used "average frequency", correspondence with the reality of what was done, however, the reference that the reviewer makes us can give more clarity to what has been done and consequently we modify "average frequency" by "breath of participation".
Point 6. Measures (p. 5; l. 231–237) – I appreciate that the authors added the description of the “Self-control” measure. However, there is no information on source of the question or its psychometric properties or its previous use (requested previously in the comment no. 12).
Response 6. In the description of the measure, a reference to the original instrument is included (Tangney et al., 2004). We hope this is what the reviewer is referring to when asking for more information on the source of the question. However, it is clear that this instrument was conceived, rather than as an evaluation of the dimensions related to self-control, as an inventory of activities related to it (Streiner, 2003). For this reason, the possibility that they were redundant was ruled out. This was corroborated with the Cronbach's Alpha measurements, which were 0.47 for the Netherlands and 0.41 for Spain and Peru.
Streiner, D. L. (2003). Being inconsistent about consistency: When coefficient alpha does and doesn't matter. Journal of personality assessment, 80(3), 217-222.
Point 7. Results (p. 8; l. 278–279) Previously I commented (no. 21) that it is not OK to stress that the Dutch showed higher rates of YAU and YBD because of higher age but it remained here without controlling for the factor of age.
Response 7. Agree, thanks, the comment was added that the analyses are performed without correction for confounders.
“Table 3 presents the results of the tests for the association between predictors and the use of YAU and YBD without correction by confounders.”
Point 8. Discussion (p. 13; l. 412+) – The authors removed the sentence about representativeness of their data but did not consider the rest of the comment (differences in questionnaire, regardless of language/semantic standardisation) or reflect the “cross-sectional” nature of data, thus using the words “risk/protective” factor even here.
Response 8. We appreciate the corrector's comment. We have changed, as much as possible, the causal expressions and have rewritten some sentences, following the reviewer's indications. For example:
“However, SL was not negatively associated with alcohol consumption in any of the countries. In addition, family leisure activities were only negatively associated with annual and binge drinking among adolescents in the Netherlands and Spain, but not in Peru. The negative association between family leisure activities and risk of UL and annual alcohol consumption (not binge drinking) was only found among Dutch adolescents with a lower level of self-control.
For both alcohol outcomes (YAU and YBD), family leisure could be considered to be a protective effect whereas unstructured leisure generally appeared to be a risk factor. (…)
Thus, in three countries within and outside Europe, the importance of young people engaging in family leisure activities was shown, while unstructured leisure activities could be considered as a risk factor for adolescent drinking. [21].
Though a higher level of self-control is associated with lower level of drinking across all three countries (except for YBD in Peru), only in the Netherlands did adolescents’ self-control influence the relation between family leisure and UL on YAU (not binge-drinking). That is, the positive character of family leisure and the risk of unstructured leisure on yearly alcohol use was strongest among Dutch adolescents with a low level of self-control.”
NEW COMMENTS:
Point 9. Introduction (p. 2; l. 62) – As I was one of the authors of this paper (ref. no. 15), I’m sure that it is an irrelevant reference in this context. We focused on organized/structured activities and well-being and not on unstructured leisure and alcohol.
Response 9. This reference is indeed not correctly cited; this study should be referred to at the end of this sentence when describing the structured (school-based) activities. This is corrected.
Point 10. Participants (p. 4; l. 184) – I do not see the point of excluding the information on two schools from the Edam-Volendam municipality (regardless of my original comment no. 9). Actually, I think it needs to be provided to the readers because it is crucial to make an objective impression of the Dutch sample.
Response 10. Thank you for the suggestion; we have added this information to the current manuscript.
“In the Netherlands, schools that offer all levels of education often have more than 1000 students. In the participating schools, 2166 adolescents filled out the questionnaire (96% response rate).”
Point 11. Measures (p. 4; l. 192–193) – As a member of the HBSC network, I have several questions regarding this new text. Why didn’t you use standardised and revised (i.e. officially approved) translations of the questions. There is a rigorous process of back-translation in place in the HBSC network, which ensures/facilitates cross-national comparability of the survey. What was the point of adapting the language version later to English? The original (i.e. source) version of all the questions included in the HBSC is English. Most crucial, I could not identify any of the HBSC questions in the Measures used in the reviewed study. The “closest” one was the YAU in the Netherlands. However, in the 2009/10 (as cited ref. no. 51) there were only 7 response options (never; 1–2 times, 3–5 times; 6–9 times; 10–19 times, 20–39 times; 40 or more times) for this particular question. Lastly, this question has not been used since 2010 and another one is in use since 2013/14 (the wording, as well as the response options have changed substantially) ... Taken the information in this comment into account, I do not find it appropriate to refer to the HBSC study at all.
Response 11. In the sentence at the top of the sources for creating our questionnaire, we misstated what we meant to say, we do not use textual questions. The HSBC questionnaire was used as a model, inspiration and reference to create our own questionnaire. Following the reviewer's suggestion, we have removed the reference to the HSBC questionnaire.
Point 12. Terminology – The authors use girls/boys here, whereas male/female distinction is present in the Tables. In the same sense, it appears to me that structured/unstructured vs. supervised/unsupervised are used interchangeably in the text, while it is not clear to me whether these are supposed to mean the same or not.
Response 12. We have corrected this deficiency by replacing the word supervised by structured and by removing the words frequency and number where we believe they are misleading.
Point 13. Measures (p. 5; l. 206) – I’m sorry. I forgot to include this comment in my original review. Why is structured mentioned twice here?
Response 13. We now see that this is not correct. We have excluded this.
Point 14. Measures (p. 5; l. 212–215) – In my former comment no. 15, I asked if the tertiles were created for a total sample or split by countries. Such a piece of information should be explicitly stated in the manuscript to facilitate its comprehension.
Response 14. Thanks for the observation. It was stated in the manuscript.
Point 15. Measures (p. 5; l. 241–242) – Although the authors indicate that the socioeconomic status was self-reported, the manuscript still mentions that this was the case only in the Netherlands and not in Spain and Peru.
Response 15. We do not know exactly what the reviewer is referring to. References to socio-economic status appear in Table 1, the socio-economic status of Spain and Peru is shown as reported by the schools, and on lines 246-250, it is stated that the socio-economic status in countries other than The Netherland was reported by the schools:
"Socioeconomic Status was self-reported in the Netherlands, and for the remaining countries it was reported by the school".
Point 16. Results (p. 7; l. 271) – The authors newly recognize that low socioeconomic status was significantly underrepresented in the sample (In the Methods section) but still keep labelling it as low in the Results.
Response 16. The results section describes the direct output of the analysis conducted for this specific sample investigated this research question. Therefore, in this sample the labelling of ‘low’ can be made, yet this has no absolute meaning. In the current version of the paper we have changed this into ‘lower’ as it makes clearer that it refers to a relatively lower level of SES.
Point 17. Results (p. 10; l. 329–331) – I do not understand the sentence. What does “less the unity” mean?
Response 17. An OR above or below the value one implies that a factor can be associated negatively (risk) or positively (protection). It should be noted that although they were not significant OR, being consistently below unity cannot be attributed to a random factor. Added a sentence at the end of the paragraph to clarify this being mentioned.
“An interesting result is that the estimates of the ORs associated with structured leisure were less the unity, although without reaching statistically significant values, which may imply that the SL as expected has a protective factor.”
Point 18. Discussion (p. 13; l. 373) – Despite claiming that the authors avoided use of “causal language” in the Discussion, it repeatedly occurs there, with the first occurrence at the end of the first paragraph.
Response 18. We think the reviewer is referring to what he indicated previously and have corrected in point 8.
Point 19. References – There is unnecessary spacing between lines + I was unable to find ref no. 56 in the text + the link for ref no. 51 was not working on my computer.
Response 19. We appreciate your pointing out these errors. We have included reference 56 which we had deleted from the text. Regarding reference 51, we have deleted it, following the reviewer's comment 3.
Edgar Benítez
